# The Effects of Laparoscopic Sleeve Gastrectomy on the Parameters of Leptin Resistance in Obesity

**DOI:** 10.3390/biom9100533

**Published:** 2019-09-25

**Authors:** Tagleb S. Mazahreh, Mahmoud Alfaqih, Rami Saadeh, Nabil A. Al-Zoubi, Moad Hatamleh, Ahmad Alqudah, Abdelwahab J. Aleshawi, Abdallah Alzoubi

**Affiliations:** 1Department of General Surgery and Urology, Faculty of Medicine, Jordan University of Science and Technology, Irbid 22110, Jordan; nazoubi@just.edu.jo (N.A.A.-Z.); moad_ihatamleh1990@hotmail.com (M.H.); abdelwahhabjamal@yahoo.com (A.J.A.); 2Department of Physiology and Biochemistry, Faculty of Medicine, Jordan University of Science and Technology, Irbid 22110, Jordan; maalfaqih@just.edu.jo (M.A.); Ah_qudah90@hotmail.com (A.A.); 3Department of Public Health and Community Medicine, Faculty of Medicine, Jordan University of Science and Technology, Irbid 22110, Jordan; rasaadeh@just.edu.jo; 4Department of Pharmacology, Faculty of Medicine, Jordan University of Science and Technology, Irbid 22110, Jordan; aaalzoubi28@just.edu.jo

**Keywords:** obesity, leptin, laparoscopic sleeve gastrectomy, body mass index

## Abstract

Obesity is a growing public health problem worldwide. Bariatric surgical procedures achieve the most sustainable and efficacious outcomes in the treatment of morbid obesity. However, little is known about the underlying molecular pathways modulated by these surgical interventions. Since leptin resistance is implicated in the pathogenesis of obesity, we herein report the effects of laparoscopic sleeve gastrectomy (LSG) on the serum levels of leptin and leptin receptor, in addition to its overall effect on leptin resistance. This was an interventional and follow-up clinical study. In the first part, patients attending the general surgery outpatient clinics at our university hospital were first stratified according to their Body-Mass Index (BMI) into cases (*n* = 38) with BMI ≥ 35 who were scheduled to undergo LSG, and controls (*n* = 75) with a normal BMI. Serum leptin and leptin receptor levels were measured by sandwich ELISA technique. A leptin resistance index was estimated by adjusting leptin to BMI ratio to leptin receptor concentration. In the second part of the study, cases who underwent LSG were followed up one year postoperatively to assess their BMI and serum leptin and leptin receptor levels. Leptin to BMI ratio was significantly higher, while serum leptin receptor was significantly lower, in obese patients compared to controls. This translated into a significantly higher leptin resistance index in obese patients. LSG resulted in a significant reduction of BMI, leptin to BMI ratio, and leptin resistance index, as it significantly increased leptin receptor levels. In conclusion, LSG showed significant decrease in leptin resistance in obese patients after one year. Further studies are needed to determine the clinical impact of this finding on LSG outcomes.

## 1. Introduction

Obesity, visceral fat accumulation, is strongly linked to the development of atherosclerosis [1], hypertension [2], and type 2 diabetes mellitus [3]. The prevalence of obesity has increased worldwide over the past four decades reaching pandemic proportions [4]. In the USA alone, it is estimated that 35% of the population are obese [5]. Globally, in 2015, 2.3 billion adults were overweight while 700 million were obese [6]. With current trends, it is estimated that around half of the adult world population will be overweight or obese by the year 2030 [6]. In addition to its global impact on public health, obesity is also an economic burden worldwide. Indeed, recent estimates indicated that the expenditure of treating obesity-related co-morbidities has reached an astounding figure of two trillion US dollars; an equivalent of 2.8% of the global gross domestic product [7]. 

Proper intervention measures to prevent or treat obesity require a thorough understanding of its etiology and risk factors. Unfortunately, the risk factors of obesity are multifactorial, intertwined, and encompass a wide spectrum of environmental, socioeconomic, and genetic components [8]. Genetic variations, for example, are associated with a 40–70% increased risk of developing obesity [9]. More than 100 genetic loci in humans are implicated in the regulation of body weight and were reported by different investigators to contribute to the obesity phenotype [9]. However, only a few genes are linked with the development of the monogenic form of obesity and these genes include, among others, *leptin* [10], *leptin receptor* [10], *proopiomelanocortin* [11], and *prohormone convertase I* [12].

Of the above list of genes, *leptin* has received much interest. The *Leptin* gene was first identified by positional cloning of the genetic locus mutated in ob/ob mice [13]. At birth, ob/ob mice are indistinguishable from their unaffected littermates. However, these mice rapidly gain weight and eventually become three times heavier than the wild type mice [13]. It was later found that ob/ob mice lose weight through daily injections with the leptin gene product [14]. Further studies have demonstrated that the leptin gene codes for a hormone secreted by fat cells [15]. Leptin belongs to a class of hormones known as adipocytokines that play an important role in regulating energy metabolism [16]. For instance, leptin secretion is increased in the well-fed state characterized by a high insulin to glucagon ratio [17]. Upon its secretion, leptin suppresses appetite and increases the metabolic rate through binding to leptin receptors expressed in the hypothalamus [18,19].

Given the aforementioned discussion, it is not surprising that loss of function mutations in the gene that codes for leptin or its receptor leads to severe early onset obesity in affected humans [20,21]. However, multiple research groups have revealed that plasma leptin levels in obese humans are usually normal or slightly elevated relative to their adipose tissue mass [22,23]. This observation suggests that resistance to leptin, rather than its deficiency, occurs in most cases of human obesity and that reducing leptin resistance rather than a mere increase in its plasma levels may reduce adipose tissue mass and obesity [23].

Several strategies are in place for the treatment of obesity. Examples are (a) lifestyle modifications with dietary, exercise, and behavioral components [24], (b) pharmacological interventions [25], and (c) surgical interventions [26,27]. Noteworthy, lifestyle modifications suffer from problems related to difficulties in compliance among obese patients and often fail to achieve sustainable weight loss [28]. Likewise, the effect of isolated pharmacological therapies is minimal and their use fails to achieve sufficient effect sizes [29]. To the contrary, results obtained from bariatric surgery appear to be most sustainable and efficacious, as supported by several prospective longitudinal studies [30]. The reason behind why bariatric surgery achieves sustained results in comparison with other strategies could be related to the effects of bariatric surgery on the gut microbiome [31], brain-gut axis [32], or the rate of gastric emptying [33]. Nonetheless, this remains a rich area of investigation. 

In this study, using an interventional cohort design of patients scheduled to receive LSG, we investigated the effects of this surgical intervention on the serum levels of leptin and leptin receptor. Our findings are further discussed in the context of how bariatric surgery affects leptin resistance. 

## 2. Subjects and Methods

The design of this study, and the clinical and biochemical endpoints were all approved by the Institutional Review Board at Jordan University of Science and Technology and conducted in accordance with the Declaration of Helsinki. Patients attending the general surgery outpatient clinics at King Abdullah University Hospital between January 2017 and July 2017 were invited to participate in the study and were formally consented upon enrollment. In the first part of the study, a case-control design was adopted. Controls (n = 75) were patients with a normal Body-Mass Index (BMI) referred for any clinical indication, and cases (n = 38) were morbidly obese patients (BMI ≥ 40), or cases with BMI ≥ 35 with obesity-related comorbidities. scheduled to undergo LSG. Demographic, anthropometric, and clinical data, in addition to baseline serum leptin and leptin receptor levels, were obtained for all participants. In the second part of the study, patients who underwent the LSG procedure were followed up one year postoperatively to assess their BMI, and serum leptin and leptin receptor levels.

### 2.1. Surgery

LSG operations were performed by the same consultant surgeon (TM). All operations followed the same procedural guidelines. Briefly, the gastrocolic omentum was divided, starting 4 cm proximal to the pylorus up to the angle of His. Dissection was performed up to the left crus of the hiatus, and all attachments were released to completely mobilize the fundus. The gastric sleeve was created using sequential firings of a 60 mm linear stapling device. The staplers were applied alongside a 36-Fr calibrating bougie positioned in the stomach against the lesser curve. The resected specimen was ultimately retrieved via the 15 mm surgical port. 

### 2.2. Measurement of Serum Leptin and Leptin Receptor Levels

Assessment of serum leptin and leptin receptor levels was performed using sandwich ELISA technique on a morning fasting venous blood sample, according to manufacturer’s recommendations. Human ELISA kits for leptin (Mybiosource USA MBS020274, minimum detection level of 10 pg/mL), and leptin receptor (Mybiosource USA MBS2883720, minimum detection level of 0.32 ng/mL) were used in this study. Serum leptin concentrations were adjusted to individual BMI readings to obtain the leptin to BMI ratio. This ratio was further adjusted to the corresponding leptin receptor concentration as a putative index of leptin resistance.

### 2.3. Statistical Analysis

Data were analyzed using the GraphPad Prism 5 software (GraphPad Software, Inc., San Diego, CA, USA), or the SAS/STAT software (version 9.3; Cary, NC, USA). Descriptive summary statistics were used to report demographic, anthropometric, and clinical characteristics of participants. Unpaired Student *t*-test was used when comparing continuous variables between cases and controls, while paired Student *t*-test was used when comparing continuous variables before and after surgery. Spearman Approximation test was used to examine the effectiveness of pairing of pre- and post-surgery readings. To adjust for confounding factors, i.e., gender and age, two statistical models were used. In model one, logistic regression was performed, considering comorbid diseases (smoking, hypertension, diabetes, and thyroid disorders) as independent variables, while pre-post-surgery differences in leptin/BMI ratio and leptin receptor concentration were considered categorical dependent variables. In model two, a linear regression model was applied, assuming that leptin/BMI ratio and leptin receptor concentrations are continuous variables. Statistical significance was set at *p* < 0.05. 

## 3. Results

### 3.1. Parameters of Leptin Resistance are Different Between Normal and Obese Individuals

Table 1 summarizes the major demographic, clinical, and anthropometric measures for all participants. It must be noted that age and gender matching was lacking in the design of our clinical study, because eligibility criteria of enrollment were solely based on BMI categorization. Diabetes mellitus or hypothyroidism was evident in 18% and 13% of obese patients, respectively (Table 1). 

We first explored whether parameters of leptin resistance are different between normal and obese individuals scheduled to undergo LSG. It was observed that serum leptin to BMI ratio was significantly higher in obese patients compared to subjects with normal BMI (*p* < 0.0001; confidence interval (CI): 38.5–45.6; Figure 1A).

Table 2 shows a summary of descriptive statistics of measured serum leptin and leptin receptor in all subjects.

### 3.2. LSG Improves Parameters of Leptin Resistance in Obese Patients

To determine the effects of LSG on parameters of leptin resistance, we compared pre-surgical levels of leptin to BMI ratio, serum leptin receptor, and leptin resistance index to those levels assessed one year postoperatively. LSG resulted in a significant reduction in BMI (*p* < 0.0001). This change was accompanied by a significant decrease in leptin to BMI ratio (*p* < 0.0001; CI: 7.1–21.9), and a significant increase in serum leptin receptor concentration (*p* < 0.0006; CI: –1.3–0.4). Moreover, leptin resistance index was markedly reduced postoperatively (*p* < 0.001; CI: 10.9–38.0; Figure 2B). These postoperative changes in leptin to BMI ratio and serum leptin receptor concentration remain significant after adjustment for confounding factors, such as age and gender (data not shown). Noteworthy, all individuals showed a decrease in leptin and BMI levels postoperatively, except for one patient who showed a slight increase in leptin level but a decrease in BMI postoperatively.

## 4. Discussion

The findings of this study provide a link between reduction in BMI achieved following bariatric surgical procedures and several parameters related to leptin resistance in obese patients. These findings may suggest evidence that changes in leptin resistance could in part explain how bariatric surgeries including LSG achieve efficacious obesity treatment outcomes. In this study, we first demonstrated that leptin to BMI ratios were significantly elevated in obese patients scheduled to undergo LSG relative to control group with normal BMI. Interestingly, the levels of serum leptin receptor were significantly lower in the obese patients. Both above parameters were combined in an index that described leptin resistance. Mathematically, this index was calculated by adjusting the leptin to BMI ratio to the corresponding levels of serum leptin receptor. Using the above mathematical expression, we demonstrated that leptin resistance was significantly elevated in obese patients compared to control individuals. Although the above findings were previously reported in other populations [34,35], this is the first study to evaluate the effect of LSG on serum leptin, leptin receptor, and leptin resistance. Leptin is one of the adipocytokines secreted by fat cells of adipose tissue and plays a major role in the regulation of energy metabolism [15]. One of the mechanisms by which leptin achieves this role is through its central control of appetite, specifically in the hypothalamus [18]. Indeed, via binding to leptin receptors expressed in cell membranes of hypothalamic tissues, leptin stimulates a signaling pathway that eventually results in the suppression of appetite [18]. Not surprisingly, inactivating mutations in leptin [21] or its receptor [20] are a cause of monogenic forms of obesity. However, similar to several other reports, we showed in the first phase of our study that levels of serum leptin are elevated in obese individuals. This finding is counterintuitive considering the effect leptin has on appetite and suggests that obesity is accompanied by resistance to leptin activity rather than a mass reduction in its serum levels [36]. Our findings further support this mechanism.

Theoretically, leptin resistance could be mechanistically achieved by lowering the mass numbers of leptin receptors expressed in target tissues or by downregulating leptin receptor activity. In this investigation, we provide evidence that leptin resistance could be caused by a downregulation in the expression of leptin receptor in target tissues since we demonstrated a decrease in leptin receptor levels in the serum of obese individuals compared to lean controls. It would be interesting to examine if leptin receptor signaling is dysregulated in obese patients of our population. This, however, requires mechanistic in vitro studies that evaluate several downstream targets of leptin receptor signaling (i.e., JAK2 and STAT3) [37]. These studies can be performed on primary cells recovered from our patients. However, it is beyond the scope of this investigation.

In the second part of this investigation, we showed that a reduction in BMI following bariatric surgery is associated with a reduction in serum leptin levels and an increase in serum leptin receptors. Several studies have reported that bariatric surgical procedures are more efficacious and provide long lasting outcomes relative to other measures used to control obesity, such as lifestyle modifications and pharmacological interventions [28,29,30]. The reason behind this observation remains unknown. Considering our current results, we propose that the effect of bariatric surgery on leptin resistance could explain the efficacy and sustainability of these procedures. Indeed, based on our findings, it could be postulated that feedback loops that regulate the levels or activity of leptin and its receptor are dysregulated in obese patients. Bariatric surgical procedures could restore these feedback loops resulting in the improvement of leptin resistance by downregulating leptin expression and/or upregulating leptin receptor levels/activity. Of note, several groups reported that bariatric surgery improves parameters of insulin resistance in obese patients [38,39]. Cross talk between insulin and leptin signaling is well documented [40,41]. Therefore, the effect of bariatric surgery on leptin resistance could at least be, in part, mediated indirectly through modulation of insulin signaling. 

Literature alludes to several difficulties in conceptualizing the term “leptin resistance”, its temporal effects on obesity, as well as confirmatory laboratory tools to measure it in patients. This indeed pertains to the fact that most studies in the field were performed on laboratory animals, dissecting the dysregulated leptin signaling pathways, and lumping them together as a potential “cellular leptin resistance”. However, such approach unfortunately cannot be extended to clinical settings. Thus, we modeled here for leptin resistance in patients by considering its three major players: BMI, serum leptin, and serum leptin receptor concentrations. Admittedly, the accuracy and clinical relevance of this novel tool need to be further investigated in larger and longer clinical cohorts.

This study is not without limitations. Firstly, it was difficult to determine based on our study design whether the improvement in leptin resistance resulted in the reduction in BMI observed in the patients or if these changes (i.e., leptin resistance parameters) were an outcome of weight reduction triggered by the bariatric surgery through other mechanisms, such as the effect on the microbiome. Secondly, we investigated the serum leptin receptors, which could not reflect the leptin receptors in tissue. Other limitations include the small sample size, the lack of reporting of other variables known to affect energy metabolism and appetite such as ghrelin and insulin hormone levels, and the non-utilization of waist circumference as a marker for obesity. 

## 5. Conclusions

LSG showed a significant decrease in leptin resistance in obese patients after one year. Further studies are warranted to determine the clinical impact of this finding on LSG outcomes.

## Figures and Tables

**Figure 1 biomolecules-09-00533-f001:**
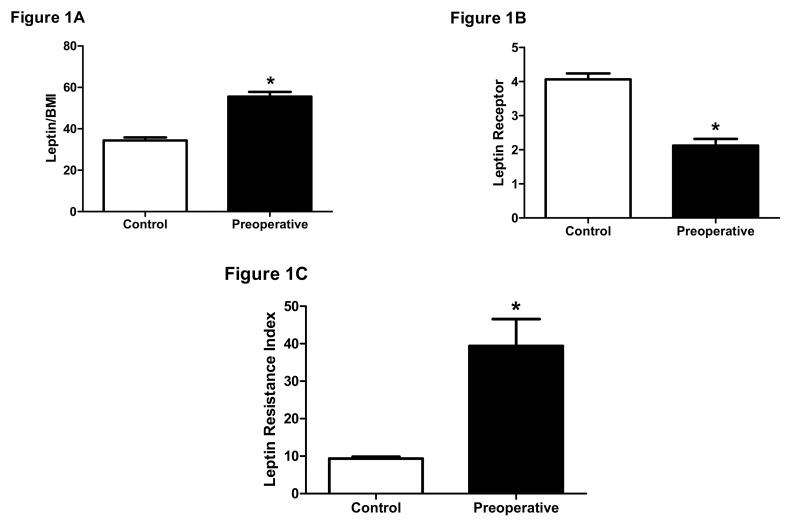
Comparison between controls and obese patients preoperatively. (**A**) Comparison between controls with normal Body-Mass Index (BMI) and obese patients preoperatively in serum leptin/BMI. * *p* < 0.05 vs. control; CI: 38.5–45.6. On the other hand, serum leptin receptor concentration was markedly decreased in obese patients (*p* < 0.0001; CI: 3.1–3.7. (**B**) Comparison between controls with normal Body-Mass Index (BMI) and obese patients preoperatively in serum leptin receptor levels. * *p* < 0.05 vs. control; CI: 3.1–3.7. Due to the lack of a definitive measure of leptin resistance in relevant literature, we opted to adjust the leptin to BMI ratio to the corresponding leptin receptor concentration as a putative index of leptin resistance. Our findings showed that obese individuals have a significantly higher index of leptin resistance compared to controls (*p* < 0.0001; CI: 14.2–24.7. (**C**) Comparison between controls with normal Body-Mass Index (BMI) and obese patients preoperatively in leptin resistance index. * *p* < 0.05 vs. control; CI: 14.2–24.7.

**Figure 2 biomolecules-09-00533-f002:**
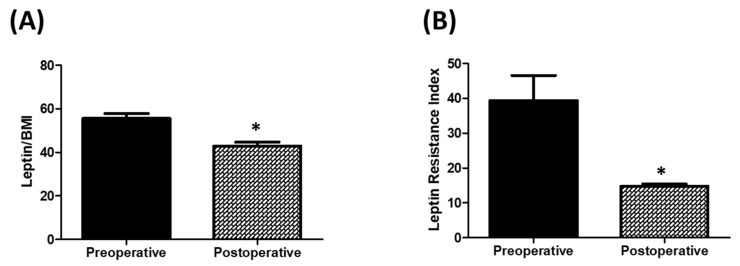
(**A**) Comparison between obese patients preoperatively and 1-year postoperatively in serum leptin receptor levels. * *p* < 0.05 vs. preoperative value; CI: 861.4–1264.8. (**B**) Comparison between obese patients preoperatively and 1-year postoperatively in leptin resistance index. * *p* < 0.05 vs. preoperative value; CI: 10.9–38.0.

**Table 1 biomolecules-09-00533-t001:** Descriptive statistics of major demographic, clinical, and anthropometric characteristics of participants.

		Cases	
	Controls	Preoperative	Postoperative
Number	75	38	
Age (Mean ± SD)	23.05 ± 2.44	37.26 ± 11.29	
Gender Male (%) Female (%)	45 (60) 30 (40)	10 (26) 28 (74)	
Comorbidity Diabetes Mellitus (%) Hypothyroidism (%)	0 (0) 0 (0)	7 (18) 5 (13)	
BMI (Mean ± SD)	22.43 ± 0.56	43.00 ± 0.73	30.82 ± 0.73

**Table 2 biomolecules-09-00533-t002:** Summary descriptive statistics of measured serum leptin and leptin receptor in all subjects.

		Cases	
	Controls	Preoperative	Postoperative
**Serum leptin concentrations (pg/mL)**			
Minimum	219.0	1196	905.0
Median	764.0	2294	1282
Maximum	1701	3314	1942
Mean	772.7	2369	1306
Std. Deviation	291.5	566.3	292.6
Std. Error	33.67	91.86	47.47
**Serum leptin receptor concentrations (ng/mL)**			
Minimum	1.600	0.3000	2.200
Median	4.000	2.050	2.900
Maximum	8.300	5.900	3.900
Mean	4.063	2.121	2.942
Std. Deviation	1.531	1.213	0.4494
Std. Error	0.1767	0.1968	0.07290

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
