# Peer review of "The Effects of Laparoscopic Sleeve Gastrectomy on the Parameters of Leptin Resistance in Obesity"

_biomolecules, 2019, doi:10.3390/biom9100533_

Round 1
Reviewer 1 Report
Mazahreh et al. present an interesting study intended to investigate the relation between several leptin markers in healthy controls and before and after sleeve gastrectomy. Several studies about leptin in Bypass cohorts has been previously published, however the sleeve gastrectomy is a novel group of patients. Therefore, the study does show novelty and is of interest in the field of bariatric surgery.
However, there are several (major) concerns about the study that need to be addressed before the study can be considered for publication.
1) The leptin levels are based upon BMI. Overall, it is known that BMI is not a sensitive marker for obesity. Especially since the baseline characteristics between controls and patients differ tremendously: BMI can be easily misinterpreted, in particular in women and women of advancing age. The results should be adjusted for at least gender, favorably also for age. Or, other markers for obesity should be used, for instance other serum markers or waist circumference, metabolic syndrome etc etc. Also, diabetes is known for changes in leptin levels as well.
2) Statistics are far from sufficient. A student's T-test for such data is not enough for showing significance. Leptin changes of the same patient before and after surgery are of interest, so at least related sample analysis should be performed, and adjustments should be done as mentioned in concern 1. Are there patients that individually do not show a decrease in leptin levels and / or BMI reduction? Could you identify why not? Or do they all show the same trend?
3) The usefullness of the leptin resistance index is unkown. The leptin and leptin receptor concentrations are of interest, however the index is unknown for any clinical significance. In my opinion, this index does not change the clinical value of the study and should be considered to be deleted from it.
4) it is unfortunate that no other markers for obesity or effect of sleeve gastrectomy are examined. From these data, it can not be said whether the effect of leptin levels is partly or completely due to reduction of BMI since calculations include BMI.
5) Via a quick search on leptin levels in other studies, the leptin concentrations are already quite high in the control groups, and especially in the obesity group. Do the authors have an explanation for this? Perhaps the method of analysis of leptin?
6) In the discussion, the authors mention that based on these data, presurgical leptin resistance could be an criteria for the positive outcome after sleeve gastrectomy. However, these data only show that the patients do, on average, show a decrease in leptin reduction after surgery and the authors did not demonstratie that patients with normal leptin levels do not respond to surgery. Again, pair-wise analysis should be done to see whether there are patients whom do not show a reduction in leptin levels. Therefore, nuance this sentence.
Minor concerns:
1) Figure 2A and 2C are obsolete, since these data are already mentioned in the tables.
2) In the discussion, authors mention that "Several studies reported that bariatric surgical procedures are more efficacious and provide long lasting outcomes relative to other measures used to control obesity, such as life style modifications and pharmacological interventions". References should be included.
3) For standardization, WHO classification of obesity should be used for the classification of patients with obesity.
Reviewer 2 Report
The manuscript entitled "The effects laparoscopic sleeve gastrectomy on parameteres of leptin resistence in morbid obesity" is very interesting.
Minor revision:
Put the Fig 2A, Fig 2B, Fig 2C, Fig 2D in one figure (Fig 2; A, B, C , D).
Reviewer 3 Report
Morbid Obesity take it of from the title or change-it.
BMI over 35 kg/mq in presence of co-morbidity ???
Morbidly obesity is defined as BMI higher then 40 kg/mq.
This was a two-phase prospective clinical study ??? It is interventional and follow up
Introduction chapter:
- line 42: Obesity, excessive fat accumulation “visceral fat accumulation”
- line 45: In the USA alone, it is estimated that 35% of the population are “is” obese
Authors should review grammar is, are, where, etc.
Chapter 2 (Subjects and Methods)
-line 91: The design “of the study”, and “the” clinical and biochemical endpoints of this study were all approved by ….
-line 97: morbidly obese patients (BMI ≥ 35) scheduled to undergo LSG.
!!!!!! Morbidly obesity is defined as BMI higher then 40 kg/mq.
!!!!! Bariatric surgery is recommended in subjects with BMI higher than 40 kg/mq or higher than 35 kg/mq at risk (presence of comorbidities).
Results chapter:
Authors must refer the confidence interval.
Discussion chapter:
-line 223, 224, 225, 226 and 227: Thake it of from the paper or summarized as future research.
Round 2
Reviewer 1 Report
The revisions of Mazahreh on the manuscript are thorough, and attempts are made to improve the concept and the statistics of the study. Although it is a small study, solely focusing on leptin measurements, it does show some new insights into the role of leptin before and after bariatric surgery.
After carefully reviewing the manuscript, I still detect minor details that need to be considered. After considering these revisions, an acceptation for publication of this manuscript is in place.
Minor details:
Please include a statement that you have conducted the study according to patient study guidelines (for instance, CONSORT guidelines or according to the declaration of Helsinki). Additional statistics are performed on your data. Please provide the statistics used in the statistics session in your manuscript. It is unfortunate that you choose not to show these data of your additional analyses in your manuscript. No major issue, but do reconsider.Author Response
Please see the attachment

Reviewer 3 Report
better
Author Response
Thank you very much